# Current and Future Antiviral Strategies to Tackle Gastrointestinal Viral Infections

**DOI:** 10.3390/microorganisms9081599

**Published:** 2021-07-27

**Authors:** Nanci Santos-Ferreira, Jana Van Dycke, Johan Neyts, Joana Rocha-Pereira

**Affiliations:** Laboratory of Virology and Chemotherapy, Department of Microbiology, Immunology and Transplantation, Rega Institute, KU Leuven, 3000 Leuven, Belgium; nanci.ferreira@kuleuven.be (N.S.-F.); jana.vandycke@kuleuven.be (J.V.D.); johan.neyts@kuleuven.be (J.N.)

**Keywords:** norovirus, rotavirus, sapovirus, adenovirus, astrovirus, viral gastroenteritis, in vitro, in vivo, enteroids

## Abstract

Acute gastroenteritis caused by virus has a major impact on public health worldwide in terms of morbidity, mortality, and economic burden. The main culprits are rotaviruses, noroviruses, sapoviruses, astroviruses, and enteric adenoviruses. Currently, there are no antiviral drugs available for the prevention or treatment of viral gastroenteritis. Here, we describe the antivirals that were identified as having in vitro and/or in vivo activity against these viruses, originating from in silico design or library screening, natural sources or being repurposed drugs. We also highlight recent advances in model systems available for this (hard to cultivate) group of viruses, such as organoid technologies, and that will facilitate antiviral studies as well as fill some of current knowledge gaps that hamper the development of highly efficient therapies against gastroenteric viruses.

## 1. Introduction

Diarrheal diseases have an estimate of 1.7 billion episodes of acute diarrhea annually, being one of the leading causes of mortality in children up to five years of age worldwide [1,2,3,4]. Viral gastroenteritis is the main cause of such diarrhea and is caused by several viruses, including human rotaviruses (HRVs), noroviruses (HuNoVs), sapoviruses (HuSaVs), astroviruses (HAstV), and enteric adenoviruses (HAdVs) [5]. These viruses have a worldwide distribution and most commonly infect children, while HuNoV infects all age groups. Viral gastroenteritis is normally self-limiting but can be prolonged and severe in vulnerable populations—namely children, elderly, and immunocompromised individuals. Infections take place in semi-closed settings—such as schools, hospitals, nursing homes, military quarters, etc.—in which people are in close contact and thus transmission is facilitated, with HuNoV in particular being associated to large outbreaks of acute gastroenteritis. From the clinical point of view, symptomatic infections have similar symptoms with an incubation period of 1–7 days (HAdV has an incubation period of 8–10 days) and with high viral shedding in the stool for the first week that can continue for several weeks [6]. Symptoms include non-bloody diarrhea and vomiting with various degrees of dehydration, abdominal pain, general malaise, and fever [7]. Correct identification of the etiological (viral) agent requires clinical laboratory diagnosis [8,9]. Currently, there are no drugs approved by the US or European regulatory agencies for treatment of viral gastroenteric infections, limiting treatment to supportive therapy with oral rehydration salts.

Human rotaviruses (dsRNA genome, *Reoviridae*) are the most important cause of virus-related diarrhea in children under the age of five, with ~125,000–200,000 deaths each year, mostly in developing countries [10,11]. The introduction of two vaccines Rotarix (GSK Biologicals, Rixensart, Belgium) and Rotateq (Merck & CO, West Point, PA, USA), significantly reduced rotavirus (RV)-related hospitalizations in children in diverse settings worldwide since 2006 [12,13]. Although these vaccines have been implemented in the national vaccination programs of 110 countries [14], the World Health Organization (WHO) estimates that 41% of all children still lack access to RV vaccines, in particular in developing countries [15]. Moreover, vaccines have been highly effective in high-income countries but considerably less potent in low- and middle-income countries. Reasons for the lower effectiveness of RV vaccination in low-income countries are at present not fully understood but may be related to (a) malnutrition that leads to zinc deficiency and avitaminoses affecting immunity; (b) gut microbiota composition; (c) co-infections; and (d) anatomical and functional abnormalities in the small intestine (environmental enteropathy) of children living in low-income countries [16]. Also, vaccines could become ineffective as novel strains could escape immunity. Finally, children with immunodeficiencies or a history of intussusception cannot be vaccinated. Taking all of this into consideration, further research to develop antiviral therapies is still needed.

Human noroviruses [(+)ssRNA genome, *Caliciviridae*] cause 700 million infections and 219,000 deaths annually, being the second most important viral cause of childhood diarrhea [17,18]. Chronic norovirus (NoV) gastroenteritis is a common complication in immunocompromised patients (e.g., transplant recipients) that can last months and is linked to great morbidity [19,20]. HuNoVs have great genetic diversity with many genotypes infecting humans, GII.4 noroviruses are the most prevalent particularly in large outbreak settings, and linked to the highest mortality and hospitalization rates over the last years [21]. Efforts to develop a HuNoV vaccine are ongoing with the most advanced candidates being developed by Takeda, in phase IIb of clinical development [22], and Vaxart, in phase Ib [23].

Human sapoviruses [(+)ssRNA genome, *Caliciviridae*] belong to the same virus family as HuNoV but are only associated with gastroenteritis in children, with genotype GI.1 causing most human infections [24]. Human astroviruses [(+)ssRNA genome, *Astroviridae*] are well known agents of gastroenteritis in children, but have recently, besides causing gastroenteritis, also been linked to neurological disease in immunocompromised patients [25,26]. Eight distinct human genotypes (HAstV-1–8) have been identified in the 1970s, HAstV-1 infections are the most prevalent worldwide. More recently, two divergent HAstV clades were identified and cause human infection: HAstV-MLB clade and the HAstV-VA clade [27]. Human adenoviruses [dsDNA genome, *Adenoviridae*] are associated with a wide range of clinical syndromes in humans [28] with gastroenteric disease being associated with serotypes from species A, D, F, and G, mainly AdV-40 and -41.

The inability to cultivate many of these diarrhea-causing viruses in standard cultivation systems has been a major challenge to overcome, which hampers our understanding of many aspects of their biology and consequently the development of antiviral strategies. Fundamental aspects of the virus life cycle such as entry mechanisms and cellular receptors are either unknown or lack details, and virus-host interactions are poorly characterized. Due to the limited animal models available, the mechanism(s) by which these viruses induce diarrhea and vomiting in the human host are incompletely described. HRVs are known to encode an enterotoxin among its non-structural proteins (NSP4), others described to disrupt junctions allowing an influx of water and resulting in increased epithelial cell permeability [29,30]. The preponderance of poor absorption, inflammation, and secretory diarrhea is not well understood, nor is the mechanism underlying virus-induced emesis since rodents are not adequate to study this hallmark of disease as they lack the vomiting reflex. All of these viruses have also strains that replicate in animals, revealing a broad host range and potential to cross species barriers [30], highlighting the possibility of zoonotic transmission.

The recently described human intestinal enteroid (HIE) model, derived from primary intestinal tissue, has opened the door to studies of in vitro replication and even pathogenesis of HRVs [31], HAdV [32], HAstV [33], and also HuNoV [34]. Since HIE are derived from non-transformed human cells and composed by the different epithelial cell populations, they recapitulate the complexity of the human intestinal epithelium allowing a more accurate study of host-pathogen interactions [35,36]. The poor understanding of diarrhea-causing viruses and high prevalence of these infections call for improved in vitro and in vivo models to study these viruses in order to develop efficient antiviral strategies.

## 2. Model Systems to Study Viral Replication and Disease

### 2.1. In Vitro Systems

#### 2.1.1. Rotavirus

Cultivation of RV in monkey kidney (MA-104) or human intestinal-derived cell lines is possible in the presence of trypsin, with most studies using a handful of strains either of simian origin (e.g., SA11) or not currently circulating human strains (e.g., Wa). Other cell lines that can be used are Caco-2 (human colon adenocarcinoma) and HT-29 (human colorectal adenocarcinoma) cells [37,38]. The latter are interesting because their features resemble to some extent to those of the human intestinal epithelium. RV induces a cytopathogenic effect (CPE) in the three cell lines mentioned and can be scored by plaque formation. On the other hand, cultivation of RV clinical isolates is often inefficient and/or yields no CPE. In RV replication NSP2 is, together with NSP5, responsible for the formation of the viroplasms (VP), which are important for replication and encapsidation of the RV genome into previrion particles. The formation of such VPs upon RV infection can be visualized using MA-104/NSP5-EGFP cells and fluorescent microscopy [39,40,41]. 

Successful replication of laboratory and human clinical RV isolates in HIE and stem cell-derived intestinal organoids (HIO) has been achieved in the past years [31,42], highlighting a promising new in vitro model to study RV infections [43]. The available and most used models to study RV and the other diarrhea-causing viruses are described in Figure 1.

#### 2.1.2. Norovirus

Due to the inability to cultivate HuNoV in standard mammalian cell cultures, the mouse norovirus (MNV) is often used as a surrogate. MNV is similar to its human counterparts in terms of the fundamental mechanisms of replication, genome, route of infection, and environmental stability [44,45,46]. A replicon system carrying the HuNoV GI.1 genome and a neomycin gene replacing the structural proteins is available since 2006 and was used to describe the antiviral activity of ribavirin, interferon [47], protease [48], and polymerase inhibitors [40,49]. However, this system has some shortcomings. The first is that it has a relatively limited replication efficacy (when compared to hepatitis C virus (HCV) replicons for example) [50] and that selection for stable expression upon transfection of naïve cells is a lengthy process. Second, the system lacks a reporter and therefore a RT-qPCR-based readout is needed, which is more time consuming and expensive, and can thus only be used in a low-throughput setting.

Many efforts have been made to grow HuNoV in immortalized cells lines with low success [51]. Vero cells were shown to allow one replication cycle of GII.4 and GII.3, thus yielding low viral loads thus not being robust enough for antiviral research [52]. More importantly, two in vitro models were brought forward to study HuNoV replication, B-cells and the HIE model. B-cells (murine and human) can be infected with MNV and HuNoV [53]. The replication observed is modest and not readily reproducible, but it was remarkable to discover that B-cells are a target cell of HuNoV (further demonstrated in [54,55]). Interestingly, the use of unfiltered faeces as inoculum rendered a higher yield of replication, which highlighted the role of histo-blood group antigen (HBGA)-expressing enteric bacteria for HuNoV replication. The HIEs resemble human intestinal morphology and physiology and allowed the in vitro cultivation of a broader group of HuNoVs strains from clinical samples [34]. Differentiated HIEs derived from different segments of small-intestine could be used successfully [34,56,57,58]. Efficient HuNoV replication in HIOs has also been described [59].

#### 2.1.3. Sapovirus

Due to the fact that until very recently there was no in vitro replication system available to study HuSaV, the porcine SaV (PSaV) Cowden strain has been used as a surrogate [60,61]. Growth of PSaV in porcine kidney LLC-PK1 cells in the presence of bile acids or porcine intestinal content is possible after passage of the virus in gnotobiotic pigs and primary porcine kidney cells. Amino acid substitutions in VP1 have been described as essential for PSaV susceptibility to replicate in LLC-PK1 cells [62]. Takagi et al. were able to replicate HuSaV in two human cell lines originated from testis and duodenum with external supplementation of bile acids [63].

#### 2.1.4. Astrovirus

The classical HAstV strains replicate in a variety of cells lines, including Caco-2, HT-29, and MA-104 cells [64]; no conventional mammalian cell culture system has been identified for the non-canonical HAstV-MLB and HAstV-VA clades. The HIE model has, however, allowed the cultivation of HAstV belonging to all three clades (classic, MLB-type, and VA-type) [33], revealing a multi-cellular tropism with VA1 infecting the various cell types present, including intestinal progenitor cells and mature enterocytes. It also demonstrated that host response to infection is dominated by interferon (IFN)-mediated innate immune responses [33].

#### 2.1.5. Adenovirus

Many human adenovirus (HAdVs) serotypes can be cultured in A549 cells [65,66]. HAdV-F serotypes, including type 40 and 41, have limited ability to replicate in this or other standard transformed cell lines [67]. However, HAdV-41 is able to replicate in HEK 293 cells [68]. The HIE model has allowed successful replication of prototype strains and clinical isolates of enteric and non-enteric HAdVs, including the enteric HAdV-41p, and allowed the discovery of new facets of HAdV biology including the sensitivity to type I and III IFNs [32]. 

### 2.2. In Vivo Systems

#### 2.2.1. Rotavirus

Previously, large animals like calves, piglets, neonatal rhesus monkeys, and lambs, were used in animal models to study RV infection [69,70,71,72,73]. Their high costs and the need for specialized facilities, equipment, and staff make them impossible to use in large scale.

Mice can be used as a small animal model to study RV replication; however, disease symptoms are only observed in new-born mice of ≤two weeks of age [74]. The underdeveloped metabolism of neonatal mice and the short period of time during which mice are susceptible to RV disease, make these mice unsuitable to test potential inhibitors of RV replication. Some efforts to develop an adult rodent model for in vivo screening of candidate anti-RV compounds have been made [40,75], including antibiotic-induced mouse microbiota depletion that allowed replication to high titers [76]. Preferably an adult rodent model should also present with clinical symptoms and allow the replication of multiple HRV strains. The use of smaller animal models is highly beneficial as for their ease of maintenance, low cost, and the ability to incorporate large numbers of animals in studies [74].

#### 2.2.2. Norovirus

Large animal models were used to assess HuNoV replication such as chimpanzees, gnotobiotic pigs, and calves [72,73,77], however, large-scale antiviral studies will hardly be feasible. The first small animal model to study HuNoV was a Rag^−/−^ γc^−/−^ BALB/c mouse model [78]. Specifically, successful replication and viral shedding of a GII.4 strain was shown, although no symptoms were observed and the infection was cleared in less than three days. Moreover, successful replication was only obtained via intraperitoneal injection; the oral route was not sufficient to cause infection because they lacked certain cell targets like Peyer’s Patches and mature M Cells [79]. Which contrasts with the fecal–oral transmission route in humans [78]. Hence studies of in vivo efficacy of candidate antivirals would be quite limited in time and disease aspects. A more robust small animal model was established by using zebrafish larvae (*Danio rerio*). Replication of HuNoV GI and various GII strains were detectable for at least six days with replication peaking at two days post infection [80,81]. HuNoV, obtained from clinical samples, was injected in the yolk of the zebrafish larvae, which is their food reservoir and thus mimicking the natural infection route. With the focus on drug discovery, zebrafish larvae are very well suited for high throughput screening due to their small size and the possibility to add the compound straight into the swimming water without a specific formulation. Yet, the molecule needs to be stable in water and the exact dose taken up by mouth, skin or gills is uncertain. Alternatively, microinjection of the compound into the pericardial cavity uses a defined amount and is suitable for compounds unstable in aqueous solution but one loses throughput [82]. This model likely constitutes the start of the zebrafish larvae as a model system in antiviral drug discovery, as it is largely unexplored in virology thus far but has great potential [83].

#### 2.2.3. Astrovirus

Currently, there is no animal model for HAstVs. A murine astrovirus model in immunodeficient mice has been reported [84], but the most widely used in vivo model are turkey poults, which are infected with the turkey astrovirus (TAstV) [29].

## 3. Antiviral Targets and Known Antivirals

Antiviral drugs could be developed to target each step of the virus life cycle—entry, replication, and release. For example, one could target the viral surface proteins thus preventing the start of infection or virion release, the viral genome replication machinery by targeting essential viral enzymes, or cellular factors to inhibit virus–host interactions. Here, we will describe the most studied antiviral compounds against the main agents of viral gastroenteritis. While for HuNoV and HRV there are antiviral compounds with known activity tested/examined in more than one system/model, that is not (always) the case with the other gastroenteric viruses. For HuSaVs, HAstVs, and enteric HAdVs, most literature available refers to antiviral research using non-human strains and/or are studied in the context of treatment of immunocompromised patients and optimization of the clinical set-up to get a good outcome.

### 3.1. Rotavirus

RV replication occurs in the mature enterocytes of the villi in the small intestine, which explains their destruction upon infection. The replication cycle starts with the attachment and cell entry mediated by VP7 and especially VP4 [85]. After cell attachment and entry [86,87], subsequent uncoating of the triple layered particle is mediated by low calcium concentration of the endosome and results in the release of the transcriptionally active double layered particle (DLP) in the cytoplasm [88]. Once these DLPs are in the cytoplasm, the VP1 proteins start the (+)mRNA transcription. This (+)mRNA serves as template for translation of viral proteins and as template for genome replication. The latter occurs in a viroplasm formed by NSP2 and NSP5 and when other specific viral proteins enter, assembly of new DLPs occur. For further maturation, which comprises removal of the transient envelope and assembly of VP4 and VP7, the DLP is budded in the endoplasmic reticulum (ER). The outer layer of the DLP consists of 260 VP6 trimers, which determine the group or subgroup of a virus strain and is essential for endogenous transcription of the genome. At last, the mature virions are released through cell lysis [85]. 

#### 3.1.1. Suppression of Virus Replication

The transcription of the RV genome can be directly inhibited by targeting the VP1, RNA dependent-RNA polymerase (RdRp), using 2′*-C-*methyl nucleosides such as 7-deaza-2′-C-methyladenosine (7DMA) and 2′-C-methylcytidine (2CMC) [40]. Brequinar (BQR) and leflunomide (LFM) are two specific dihydroorotate dehydrogenase (DHODH) inhibitors that robustly inhibited RV replication in Caco-2 cells as well as in HIEs in both laboratory strain SA11 and RV strain 2011K isolated from clinical sample. The authors hypothesized that BQR and LFM act by depleting pyrimidine nucleotide pool through targeting DHODH [89]. Anti-RV compound class, stage of life cycle where acts, molecular target, and mechanism of action is summarized in Table 1.

#### 3.1.2. Inhibition of Viroplasm Formation

ML-60218 is an indazole sulphonamide known as an inhibitor of RNA polymerase III. Eichwald et al. demonstrated in vitro that ML-60218 was able to disrupt already assembled VPs and to hamper the formation of new ones in a dose-dependent manner, resulting in a reduction in accumulated viral proteins and newly made viral genome segments, disappearance of the hyperphosphorylated isoforms of the viroplasm-resident protein NSP5, and inhibition of infectious progeny virus production. Moreover, ML-60218 was able to induce structural damage into DLPs, indicating that interferes with the formation of higher-order structures of VP6, the protein forming the DLP outer layer. Despite its potent anti-rotavirus activity, ML-60218 presented high cytotoxicity to MA-104 cells and low solubility requiring further optimization of the molecule [90].

Nitazoxanide (NTZ) is an antiprotozoal agent with an antiviral activity against a large number of viruses, including RVs and NoVs [91]. NTZ showed potent in vitro antiviral activity against SA11 and HRV G1P [8] in MA-104 cells [92]. However, a very moderate anti-RV effect was observed with 25 and 50 μM of tizoxanide at 12 hpi in RV-ST3-infected cells and was not able to reduce RV-induced CPE [40]. Studies have shown that tizoxanide inhibits the maturation of RV VP7, a glycoprotein that forms the outer part of the virion and one of the six structural glycoproteins involved in RV replication, alters viroplasm formation and interferes with viral morphogenesis [92]. It underwent several phase 2 clinical trials where NTZ showed to significantly reduce the duration of symptoms in adults, adolescents, and children infected with RV and NoV [93,94].

Ursolic acid (UA) is a natural pentacyclic triterpenoid that showed potent anti-RV activity in vitro. Since UA is a hypolipidemic agent, the authors hypothesize that the anti-RV activity is mediated by the decrease in the availability of lipid droplets (ER-derived intracellular organelles for neutral lipid storage) that are required for VP formation. Several modifications were introduced to increase pharmacokinetic profile (due to low bioavailability) but none of them has yet shown anti-RV activity [95].

#### 3.1.3. Targeting Host Cell Factors Essential for Viral Replication and Others

Infected cells activate innate immunity mechanisms to reach an antiviral status through the synthesis and secretion of high levels of IFNs in response to the presence of viral RNA. RV has been demonstrated to evade this immune response by inhibiting the production of IFNs [96]. Therefore, maintaining high IFN levels in infected cells may be an effective anti-RV strategy [97]. Cyclosporine (CsA) is a calcineurin inhibitor widely used as immunosuppressant agent that also can inhibit RV replication in vitro and in vivo. Moreover, it restored IFN-β expression in RV-infected HT-29 cells and in a RV-infected neonatal mouse model, suggesting that CsA modulates the expression of key regulators in IFN signaling pathway, promoting type I IFN-based intracellular innate immunity in RV host cells [98]. Similar results were found with cordycepin, an adenosine analogue that reduce propagation of different RV strains in vitro and murine strain in BALB/c mice [99].

18-β-Glycyrrhetinic acid (18βGRA) is an aglycone and the active metabolite of glycyrrhizin after gut commensal metabolization which inhibited RV replication in both in vitro and in C57BL/6 mice [100,101]. Even though the anti-RV mechanism is not totally clear, the authors hypothesized that the effects of 18βGRA might be due to its capacity to modulate the PI3K/Akt pathway [101].

Further antiviral compounds targeting other RV components—e.g., gemcitabine [102] and 6-thioguanine [103]—have been reviewed elsewhere [97,104]. Due to limited in vitro results, more research is necessary to characterize their antiviral action.

### 3.2. Norovirus

The current knowledge on HuNoV replication still derives partly from studies with related caliciviruses and is based on the analogy with other (+)ssRNA viruses, due to the fact that until recently there was a lack of robust in vitro and in vivo cultivation systems to study HuNoV replication. While many questions remain about the replication mechanisms of HuNoV, this is in contrast to the better understood MNV replication process. An important breakthrough was the identification of proteinaceous receptors (CD34, CD300lf, CD300ld) that modulate and facilitate MNV entry and infection [105,106,107]. Although the process of HuNoV uncoating is not known, recent work showed that the minor capsid protein of feline calicivirus (FCV) forms a pore in the capsid upon receptor engagement, putatively playing an important role in viral genome release [108]. The translation mechanism, used by all viruses of the *Caliciviridae* family, is characterized by direct VPg-mediated recruitment of the eukaryotic translation initiation machinery [109]. Translation of the ORF1 results in the release of all the non-structural proteins which then initiate the formation of the replication complex [110,111,112]. Within this replication complex, the RdRp uses the mRNA template for the synthesis of the (-) ssRNA intermediate resulting in the creation of a double stranded replicative form. The RdRp uses the uridylylated VPg as a primer and with the newly synthesized RNA intermediate it produces new positive-sense genomic and subgenomic RNA [113]. The RdRp has emerged as an optimal target for the development of antiviral drugs since it plays a pivotal role in viral replication [114,115]. In the final step, the newly synthetized genomes are packaged into virus capsids formed by the two structural proteins VP1 and VP2, followed by virion assembly and exit.

#### 3.2.1. Targeting Virus Binding to Host Cell Surface

In order to attach to the cell surface, the P2 subdomain of the VP1 capsid protein interacts with the HBGAs, heparan sulphate or sialic acid [116,117,118,119]. HBGAs are thought to facilitate HuNoV attachment and entry whilst sialic acids facilitate entry of MNV. However, the exact mechanism behind the HBGA-HuNoV interaction remains unknown, as there are HuNoV strains that do not interact with any of the available synthetic HBGAs [120,121]. This indicates that another (still unidentified) protein receptor or additional co-factors may be required for HuNoV infection [122,123,124,125]. Several carbohydrate analogs with structures resembling fucose, such as citrate and other glucomimetics, have been described to inhibit viral capsid attachment to HBGAs [126]. These compounds have been identified by in silico and in vitro screening after the crystal structure of NoV bound to HBGAs was solved [127]. HIEs offer a first opportunity of utilizing an in vitro model to study the antiviral activity of these compounds since it has been shown that HuNoV infection is dependent on HBGAs expression in intestinal cells [34]. Anti-NV compound class, stage of life cycle, molecular target, and mechanism of action are summarized in Table 2.

#### 3.2.2. Targeting the Viral Protease

NoV 3CLpro enzyme catalyzes the cleavage of the viral polyprotein into non-structural proteins during virus replication. Rupintrivir is a protease inhibitor designed for the treatment of human rhinovirus but showed to have broad-spectrum antiviral activity against other picornaviruses, coronaviruses and caliciviruses [48,128,129,130]. Strategies to improve activity targeting the active site with other scaffolds have been pursued [131]. Furthermore, other peptidomimetic compounds that interact with the active site of protease have been designed [130].

#### 3.2.3. Targeting the Viral RdRp

The most promising group of RdRp inhibitors are nucleoside analogues, as these directly bind to the very well conserved active site of the RdRp, after conversion to their triphosphate active form, which prevents the incorporation of the next nucleotide resulting in the formation of an incomplete and nonfunctional RNA strand. Since the active site of the RdRp is also highly conserved among viral families, nucleoside analogues may have a broader spectrum activity [132]. Some of these nucleoside analogues have shown activity against multiple genotypes and in many model systems.

It comes as no surprise thus that the only small molecule antiviral that has yet moved into clinical development for the treatment of HuNoV infection is the purine nucleoside CMX521 (Chimerix Inc., Durham, NC, USA). Although a phase I trial was completed with success, its development was halted in 2018 [133]. However, little information has been published on the antiviral activity of this nucleoside; most studies utilized other nucleosides active against related viruses. The most studied compound has been 2′*-C-*methylcytidine (2CMC), a cytidine analogue that was initially developed for HCV [134]. Since it has been tested in every available model, it has been regarded as a benchmark compound for NoV [49,56,80,135,136,137,138]. Jin et al. showed that 2CMC triphosphate (2CMC-TP) inhibited the viral polymerases by competing directly with natural CTP during primer elongation, therefore acting as a classic chain terminator [139]. 7-deaza-2′*-C-*methyladenosine (7DMA) and NITD008 are adenosine analogues that were initially developed as an inhibitor of HCV and dengue virus replication, respectively, but also possess anti-NoV activity [40,140].

Another mechanism by which nucleoside-like compounds can act as antivirals is lethal mutagenesis. Although this concept was first brought forward using the broad-spectrum compound ribavirin, it has gained strength and interest from the discovery of favipiravir as a broad-spectrum antiviral. Clinical studies with this molecule have been carried out during the Ebola outbreak in West Africa in 2014–2016 and are also ongoing for COVID-19 (ClinicalTrials.gov NCT04373733, accessed on 3 June 2021) [141]. Studies with SARS-CoV-2 infected hamsters showed that favipiravir reduces viral infectivity despite having a modest effect on viral RNA loads—this is in line with lethal mutagenesis as a main mechanism. A study using the drug to treat a patient with common variable immunodeficiency suffering from a chronic HuNoV infection showed symptomatic response to favipiravir treatment, along with evidence for selective pressure on the infecting HuNoV population [142]. Hence, further studies addressing its clinical efficacy would be desirable in the context of chronic HuNoV infections.

Furthermore, biochemical studies showed that favipiravir inhibited the NoV polymerase by competing mostly with ATP and GTP at the initiation and elongation steps. Unlike the classic nucleosides, favipiravir did not cause immediate chain termination of NoV RdRp indicating that it may indeed act by multiple mechanisms of action [137,139,143,144]. Its in vitro antiviral effect was shown to be moderate and in vivo effects on MNV-infected mice yielded variable results; studies using the more recent HIEs and zebrafish models are still lacking.

Ribavirin, a guanosine analogue with broad-spectrum activity against both RNA and DNA viruses, was shown to have an anti-NV effect but this was also moderate. It was suggested that this anti-NoV effect was exerted by depletion of intracellular GTP pools and not via a direct interaction with the RdRp [47,145].

Other molecules, i.e., non-nucleoside analogues, can inhibit the RdRp by allosteric mechanisms. In silico approaches combined with RdRp enzymatic assays led to the identification of such inhibitors and to the characterization of two binding sites within the NoV RdRp [146,147]. NAF2, suramin, PPNDS, and NF023 have also been described as inhibitors of the NoV RdRp [148,149,150,151,152]. However, poor cell permeability prevented confirmation of antiviral activity in vitro [146,147,148].

#### 3.2.4. Targeting Host Cell Factors Essential for Viral Replication

IFN type I and II were shown to have an impact on NoV infections by triggering the host innate immune response [153,154,155]. Also type III IFN (IFN λ) was shown to protect mice against MNV challenge, therefore this could be explored for the treatment/prophylaxis of NoV infections [156,157].

In addition, Toll-like receptor (TLR) 7 agonists, like resiquimod (R-848), that stimulate IFN production, where shown to block MNV replication. At high concentrations R-848 also reduced replication of the HuNoV GI.1 replicon by 50% [158]. The TLR4 agonist, poly-γ-glutamic acid (γ-PGA), could also inhibit MNV replication very efficiently in vitro and in vivo [159].

Another host factor that interacts and plays a role in NoV replication is the heat shock protein 90 (Hsp90), a chaperone protein that assists in the maturation of multiple proteins. The inhibition of Hsp90 activity by 17-dimethylaminoethylamino-17-demethoxygeldanamycin (17-DMAG) resulted in the inhibition of MNV replication in vitro and in vivo [160].

#### 3.2.5. Others

The mechanism of action to which nitazoxanide restrains HuNoV infection in not yet known. Despite that, NTZ is the only HuNoV antiviral candidate to complete clinical trials, showing a reduction of the duration of symptoms [93]. However, NTZ failed to eradicate NoV infection in a chronically infected immunocompromised patient [161]. Moreover, studies into its anti-NV mechanism of action are lacking.

### 3.3. Sapovirus

Due to lack of culture systems very few antiviral studies have been performed targeting SaV infections. Since HuSaV belongs to the family *Caliciviridae*, antiviral drugs active against HuNoV are considered potentially active against this other genus of the family, particularly those targeting their most similar proteins (like the RdRp). In fact, we demonstrated in an earlier study that 2′*-C-*methyl nucleoside analogues with anti-NoV and anti-RT activity extended the antiviral activity also to SaVs [40]. Antiviral compound class, stage of life cycle where acts, molecular target and mechanism of action is summarized in Table 3.

### 3.4. Astrovirus

Classic HAstVs can be grown in immortalized cell lines. The virus replication strategy was thus shown to be similar to that of caliciviruses, i.e., via the formation of a replication complex where viral genome synthesis occurs [27]. Antiviral studies available to date used repurposed compounds such as ribavirin [162], favipiravir [163], and nitazoxanide [91] (Table 3).

#### 3.4.1. Targeting Virus RdRp

Ribavirin and favipiravir both showed activity against HAstVs. Ribavirin inhibits replication of HAstV-VA1 and classic HAstV-4 in vitro [164]. However, Hargest et al. found that ribavirin failed to inhibit HAstV-1 replication up to a concentration of 250 µM [165]. Favipiravir was able to inhibit replication of HAstV-VA1 but less efficient in HAstV-4 [164].

#### 3.4.2. Others or Unknown Target

The antiviral mechanism of action of nitazoxanide (NTZ) against AstV has not yet been established, but research suggests it may be via the induction of the IFN response due to activation of protein kinase R or disruption of the unfolded protein response [91]. Hargest et al. showed that NTZ was able to inhibit HAstV-1 replication in vitro by disrupting early events in the replication cycle, and had in vivo efficacy in a turkey AstV model [165]. Studies of the anti-astrovirus effect of NTZ in the more clinically relevant HIE model is still lacking.

### 3.5. Adenovirus

#### 3.5.1. Targeting Viral DNA Polymerase

Pyrrolopyrimidine derivatives were reported to target the AdV polymerase [166]. In addition, Mohamed et al. synthetized pyrrole and pyrrolopyrimidine derivatives (pyrrolo[2,3-d]pyrimidine and pyrrolo[3,2-e][1,2,4]triazolo[1,5-c]pyrimidine) where two compounds (7f and 12a) showed in vitro antiviral activity against HAdV type-7 [167].

Cidofovir (CDV) is a nucleotide analog with broad antiviral activity against DNA viruses including AdVs [168,169,170]. Some case reports have demonstrated success in using CDV to treat disseminated AdV in immunocompromised patients, including hematopoietic stem cell transplant patients [171] and treatment of extra-gastrointestinal infection in immunocompetent patients [172], making CDV the drug of choice for severe AdV infections [173,174]. Due to low bioavailability of CDV and its nephrotoxicity, attempts to develop derivatives with better pharmacokinetics profiles were done. Brincidofovir is a lipid-conjugated derivative of cidofovir that was shown to be activity against HAdV in vitro (including AdV serotype 31) and in vivo (non-enteric AdV) [175] (Table 3). However, phase II clinical trials shown no effect when compared to placebo treatment [176]. Other inhibitors of HAdV DNA polymerase have been described [177,178], but studies of antiviral activity against enteric HAdV species are lacking.

#### 3.5.2. Others

Several other compounds with different mechanisms of action, including gene expression and epigenetic disruptors, nuclear transport inhibitors, protease inhibitors, and CDK inhibitors, have been described as active against HAdV [179]. Most in vitro and in vivo testing have been performed with respiratory HAdV species. To determine the antiviral activity of these compounds against enteric AdV species would be of great importance to find a more suitable antiviral treatment for gastroenteric HAdV.

### 3.6. Conclusions

As the causative agent of viral diarrhea cannot be identified based on clinical symptoms, it would be highly advantageous to develop a syndrome-based treatment against viruses that cause acute gastroenteritis rather than developing an antiviral drug against a single virus. In acute infections, the availability of a single and potent antiviral treatment would allow for a faster start of treatment and would grant higher chances of success. On the other hand, such antiviral treatment would also be of great value to treat risk patients chronically infected patients, contain extensive outbreaks and to be prepared when new viral strains emerge. Therefore, a continuous commitment to the development of antivirals alongside the vaccine approach is needed.

For this approach to be possible, one can think of a single antiviral targeting a highly conserved viral protein or a combination of antivirals against the viruses in the group, possibly acting on critical steps of the life cycle of each virus. Targeting the viral RdRp is in our perspective the best starting point for either single or combination therapy [138]. With a specific target in mind, a good next step would be an in silico structure-based screening of nucleoside and non-nucleoside small-molecules, using a series of molecular docking simulations on the crystal structures or homology models of RdRps of these diarrhea-causing viruses. This strategy would allow the identification of chemical scaffolds with good predicted affinity for all proteins at the same time, while their subsequent assessment in enzymatic inhibition assays would provide potential broad-spectrum hits for further optimization. This strategy is feasible as specific active sites and possible binding sites on the RdRp are already known.

## 4. Future Perspectives and Challenges

With the numerous methodological advancements to study gastroenteric viruses over recent years, there is an increased insight into many features of their replication. These new models will help to discover novel inhibitors and to elucidate the mechanism of action of some of the previously described compounds. Thanks to the intestinal organoid system, it is now for the first time possible to not only cultivate viruses such as HuNoV but also to infect the same cells with the various agents of viral gastroenteritis. To do so in highly physiologically relevant cultures, possibly in parallel, is highly beneficial when aiming to advance significantly antiviral research for gastrointestinal disease. Moreover, the road is open to validate drug efficacy in chronic patients by a personalized medicine approach, given organoid cultures can be started after an intestinal biopsy of the patient is harvested and the available drugs tested using the patient-derived cultures.

Still, many aspects of the biology of gastroenteric viruses remain elusive, demanding a continuous refining of the available model systems and the development of new ones, which collectively can answer the remaining questions and result in the development of therapeutics. One important caveat is the impossibility to grow HuNoV virus stocks in organoid cultures thus far, which suggests replication is restricted in some manner. This is a fundamental aspect for simple and standardized high-throughput assays to be developed, which are required for successful drug discovery campaigns to take place. The coming years will likely bring major advances in terms of commercially available organ-on-chip systems, culture media and other materials necessary to push organoid-type cultures to the next level and allow them to take center stage in antiviral drug development.

## Figures and Tables

**Figure 1 microorganisms-09-01599-f001:**
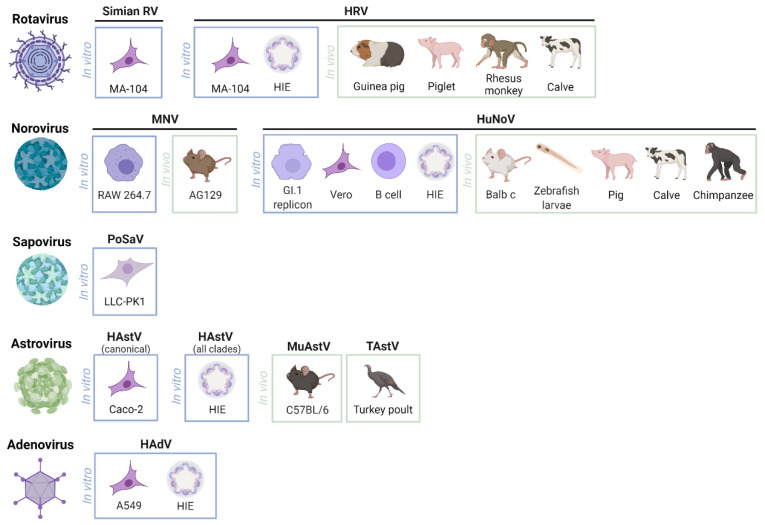
Most described in vitro and in vivo models to study human rotavirus, norovirus, sapovirus, astrovirus and enteric adenovirus, and their surrogates. RV—Rotavirus; HRV—Human rotavirus; HIE—Human intestinal enteroid; MNV—Mouse norovirus; HuNoV—Human norovirus; PoSaV—Porcine sapovirus; HAstV—Human astrovirus; MuAstV—Murine astrovirus; TAstV—Turkey astrovirus; HAdV—Human adenovirus. Created with BioRender.com.

**Table 1 microorganisms-09-01599-t001:** Antiviral compounds for rotavirus infections within this review.

Antiviral Compound	Class of Inhibitor	Stage of Viral Life Cycle	Molecular Target	Mechanism of Action
2CMC	Nucleoside analogue (cytidine)	Genome replication	RdRp	Direct inhibition of viral RdRp acting as final chain terminator
7DMA	Nucleoside analogue (adenosine)			
Brequinar	Quinolinecarboxylic acid		DHODH	Blocking de novo pyrimidine biosynthesis by inhibition of DHODH
Leflunomide	Isoxazole derivative			
ML-60218	Indazole sulfonamide	Viroplasm formation	VPs; DLPs	Disruption of VPs and hampering formation of new VPs; Induction of structural damage into DLPs hampering VP6 formation
Nitazoxanide	Thiazolide		VPs	Inhibition of VP7 maturation, hampering VP formation; Interference in viral morphogenesis
Ursolic acid	Triterpenoid		Lipid droplets	Decreases lipid droplets availability required for VP formation
Cyclosporine	Cyclic peptide	Host factor	IFN signalling pathway	Increase expression of type I IFN
Cordycepin	Adenosine analogue			
18βGRA	Aglycone		PI3K/Akt pathway	Modulation of PI3K/Akt pathway, increasing cell apoptosis and preventing virus replication

2CMC—2′-C-methylcytidine; 7DMA—7-deaza-2′-C-methyladenosine; 18βGRA—18-β-Glycyrrhetinic acid; RdRp—RNA dependent-RNA polymerase; DHODH—dihydroorotate dehydrogenase; VP—Viroplasm; DLP—Double layered particle; IFN—Interferon.

**Table 2 microorganisms-09-01599-t002:** Antiviral compounds for norovirus infections within this review.

Antiviral Compound	Class of Inhibitor	Stage of Viral Life Cycle	Molecular Target	Mechanism of Action
Citrate	Carbohydrate analogue	Viral entry	Viral capsid	Blocks binding of P domain of viral capsid to HBGAs
Rupintrivir	Peptidomimetic inhibitor	Translation	Viral protease	Inhibition of NoV 3CLpro blocking the cleavage of NS polyprotein, essential for production of viral progeny
CMX521	Purine nucleoside	Genome replication	RdRp	Direct inhibition of viral RdRp acting as final chain terminator
2CMC	Nucleoside analogue (cytidine)			
7DMA	Nucleoside analogue (adenosine)			
NITD008	Nucleoside analogue (adenosine)			
Favipiravir	Nucleoside analogue (pyrazine)			Direct inhibition of viral RdRp by competition with ATP and GTP at the initiation and elongation steps; Lethal mutagenesis
Ribavirin	Nucleoside analogue (guanosine)			Inhibition of viral RdRp by depletion of intracellular GTP pools
NAF2	Non-nucleoside analogue			Allosteric inhibition of RdRp
Suramin				
PPDS				
NF023				
Resiquimod	TLR agonist	Host factor	TLR7	Stimulation of IFN production by TLR7 agonism
γ-PGA			TLR4	
17-DMAG	-		Hsp90	Inhibition of Hsp90 activity
Nitazoxanide	Thiazolide	Other	Not known	Not known

2CMC—2′-C-methylcytidine; 7DMA—7-deaza-2′-C-methyladenosine; γ-PGA—Poly-γ-glutamic acid; 17-DMAG—17-dimethylaminoethylamino-17-demethoxygeldanamycin; RdRp—RNA dependent-RNA polymerase; TLR—Toll-like receptor; Hsp90—Heat shock protein 90; HGBGAs—Histo-blood group antigens; IFN—Interferon.

**Table 3 microorganisms-09-01599-t003:** Antiviral compounds for sapovirus, astrovirus, and enteric adenovirus infections within this review.

Virus	Antiviral Compound	Class of Inhibitor	Stage of Viral Life Cycle	Molecular Target	Mechanism of Action
SaV	2CMC	Nucleoside analogue (cytidine)	Genome replication	RdRp	Direct inhibition of viral RdRp acting as final chain terminator
	7DMA	Nucleoside analogue (adenosine)			
AstV	Ribavirin	Nucleoside analogue (guanosine)	Genome replication	RdRp	Inhibition of viral RdRp
	Favipiravir	Nucleoside analogue (pyrazine)			
	Nitazoxanide	Thiazolide	Other	Not known	Possible induction of IFN response by activation of protein kinase R
AdV	Cidofovir	Nucleoside analogue (cytosine)	Genome replication	Viral DNA polymerase	Direct inhibition of viral DNA polymerase acting as final chain terminator
	Brincidofovir	Nucleoside analogue (cytosine)			
	Compound 7f and 12a	Pyrrolopyrimidine derivatives			

SaV—Sapovirus; AstV—Astrovirus; AdV—Adenovirus; 2CMC—2′-C-methylcytidine; 7DMA—7-deaza-2′-C-methyladenosine; RdRp—RNA dependent-RNA polymerase; IFN—Interferon.

## Data Availability

All the data is present in the manuscript.

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
