# Peer review of "Current and Future Antiviral Strategies to Tackle Gastrointestinal Viral Infections"

_microorganisms, 2021, doi:10.3390/microorganisms9081599_

Round 1

Reviewer 1 Report

In their paper ”Current and future antiviral strategies to tackle gastrointestinal viral infections”, Ferreira et al. review the current literature on the antiviral strategies for controlling enteric viral diseases. This is a broad and some times very complicated topic but I think the authors have managed to write a very comprehensive and well-written review. In my opinion the article can be accepted in the current format.

Author Response

Thank you for your comment

Reviewer 2 Report

This review article is well written and provides comprehensive knowledge about the background of human enteric viruses and some useful information regarding related in vitro and in vivo models. A few comments to further improve this article:

  1. Too few details about culturing system of sapovirus. There are actually more cell lines for porcine SaV such as J Virol. 2016 Feb 1; 90(3): 1345–1358. Also, enteric Adv like serotype 40 can be propagated in HEK 293T cells (https://doi.org/10.1590/S0074-02762009000500013). The authors should do a better job in collecting related literature.
  2. Tons of antivirals that the authors mentioned act on different viruses and different stages of viral life cycle or host factors. It will be better for readers to have diagrams illustrating which drug acts on which component in the virus-host scenario.
  3. Similar to #2, a couple of tables listing the names of the specific antiviral, the targets, and the brief mechanisms will be very much appreciated by readers.
  4. The antivirals against AdV mentioned by the authors are few and lack of detailed information. There is a very good example published early this year which elaborately talked about more anti-AdV drugs, and the authors could and should take this article into consideration to improve their section 3.5. Certainly, it is crucial to focus on enteric AdV: https://www.sciencedirect.com/science/article/pii/S0166354221000243?dgcid=rss_sd_all  
  5. Lastly, the authors should keep an eye on the latest literature. A good amount of references in this review article are obsolete and require updates, which can make this review distinct from the previous ones.  
